# What Makes Consumers Purchase Mobile Apps: Evidence from Jordan

**Ahmad Samed Al-Adwan [1,*]** and **George Sammour [2,3]**

1   Electronic Business and Commerce Department, Business School, Al Ahliyya Amman University, Amman 19328, Jordan
2   Management Information Systems Department, Business School, Al Ahliyya Amman University, Amman 19328, Jordan; g.sammour@ammanu.edu.jo
3   Business Information Technology Department, Princess Sumaya University for Technology, Amman 19328, Jordan
*   Correspondence: a.adwan@ammanu.edu.jo

**Abstract:** Mobile applications (mobile Apps) have changed the ecosystem of the business world. The rapid progression in the market for smart devices and mobile Apps has brought about a revolution with regard to the mobile Apps' economy. The major revenue stream of this economy is the sale of mobile Apps, with such sales being expected to increase dramatically every year. However, in spite of this, a considerable number of mobile Apps fail to capture consumers' attention. Additionally, in developing countries such as Jordan, there is lack of research into understanding and determining the major factors that influence consumers' decisions to purchase mobile Apps. Accordingly, the main objective of this study is to identify the key factors that Jordanian consumers consider in determining whether or not to purchase mobile Apps for their smartphones. To achieve this objective, a mixed-methods approach is adopted. An exploratory study involving a qualitative methods approach (interviews with consumers) is first conducted in order to determine the factors that influence consumers' purchase behavior decisions. Then, a confirmatory study that employs a quantitative approach (a survey questionnaire) is undertaken to test the proposed model, the building of which is based on the findings of the exploratory study. The results indicate that seven factors are recognized as being pre-eminent when it comes to decisions to purchase mobile Apps. These factors include the price value of the App, App performance, App enjoyment, App trialability, electronic word-of-mouth (eWOM) about the App, App technical reliability and App usefulness. While this study advances our understanding of the main factors that influence mobile Apps purchase, it also reveals useful implications for mobile Apps publishers and developers, in order to promote and increase the sales of Apps.

**Keywords:** mobile applications; Jordan; enjoyment; price value; usefulness; performance; technical reliability; trialability

---

## 1. Introduction

The commercialization of the Internet and the rapid advancement of information technology (IT) have rapidly changed the business landscape [1]. Mobile applications (mobile Apps) have become increasingly popular in developed countries and in most developing markets due to the extraordinary growth and development of the smartphone market. Consequently, a new economic landscape, entitled the App economy, has emerged [2]. Taylor et al. [3] point out that mobile Apps are defined as " ... small programs that run on a mobile device and perform tasks ranging from banking to gaming and web browsing" (p. 60). These apps are classified into different categories such as gaming, online shopping, sport, business, social networking and book Apps [4]. As of the first quarter of 2020,

a report indicates that the top leading App Stores—Google Play and App Store—offer a total number of 2.56 million Apps and 1.85 million Apps, respectively [5]. Furthermore, as demonstrated by recent statistics, the gross App revenue in the Google Play store in 2019 hit USD 29.3 billion [6], while the gross App revenue of the App Store amounted to USD 54.2 billion [7]. The number of smartphone users worldwide is expected to reach 3.8 billion by 2021 [8], and the global smartphone App industry is also predicted to grow to USD 693 billion by 2021 [9].

While App publishers can obtain revenues from various sources (e.g., in-app advertising, in-app purchasing), App sales are viewed to be of the most important revenue stream [3]. Importantly, purchasing and using paid mobile Apps is critical for the success of App service providers and developers. While free mobile Apps are available to users at no upfront cost and do not provide in-app purchases, paid apps necessitate users to pay in advance for the initial download. Free Apps developers are relying on several methods to create revenue in addition to directly generating money from paying users. For instance, one of the main revenue streams for free mobile Apps is partnering with advertising networks in order to serve ads to users. Thus, free Apps dominate the majority of the market share as 90% of mobile Apps in the market are free, and it is expected that the revenue generated from paid Apps will dramatically decrease in the coming years [10]. Therefore, paid App publishers and developers have been actively seeking methods to boost sales and increase the revenue generated by such Apps [11]. Furthermore, along the same lines, App markets such as App Store and Google Play are employing various marketing practices such as App rankings regarding sales and popularity, online feedback tools (e.g., reviews and rating), and are introducing trial versions to increase sales. The importance of such efforts is amplified especially in light of the comparatively high costs required to develop mobile Apps and their low prices. It is suggested that the average overall price for the majority of paid Apps in the App Store and the Google Play store is less than USD 1 [12,13]. Therefore, the need to stimulate Apps sales considerably is crucial in order to recover the development costs and to ensure a profit. However, in order to increase sales, publishers need to have a comprehensive understanding of the main factors that influence customers' decisions to purchase paid mobile Apps.

In spite of the need to comprehend what consumers are expecting with the aim of increasing App sales, very limited research is available on why and how consumers decide to purchase paid mobile Apps. Specifically, most of the research in this field has primarily been directed to tackling issues of App adoption (e.g., [14,15]), App design and development (e.g., [16,17]), and App usage (e.g., [18,19]). It is worth noting that mobile App adoption is, to some extent, different from App purchase. Specifically, consumers who purchase paid Apps (the focus of this research) have double roles, as they act as technology adopters and purchasers. For instance, such consumers are required to purchase the paid Apps prior to adopting and using them. Certainly, personal use is viewed as the main motivation behind purchasing Apps, whereby individuals accept the cost of such voluntary purchase [2]. This case differs from the adoption of free systems or Apps or that have been offered to users by an organization. From this perspective, it is argued that there is a lack of models and frameworks to explain decision making with regard to mobile App purchase. Previous research is limited in terms of studies that offer explanations of decision-making factors with respect to App purchases that can result in generating revenue for mobile Apps publishers and developers. Such a gap existing in the literature, as well as the important practical issue this signifies, offers the incentive for undertaking the current study.

## 2. Study Aim and Significance

The key aim of this study is to explore the major factors underpinning consumers' decisions to purchase mobile Apps. In order to achieve this goal, a mixed methods approach is adopted. Specifically, this approach consists of an exploratory study (a qualitative study) accompanied by a confirmatory one (a quantitative study). The use of a mixed method approach helps in overcoming the act of selecting a set of factors that have already been empirically examined as factors that impact mobile

App purchasing intentions, and thereby results in misleading and irrelevant findings. Furthermore, this study evades the use of popular theories and models used to examine mobile App adoption, as such, theories and models may not be relevant.

Xu and Khazanchi [20] stress that mixed methods approaches have gained a wide popularity in social sciences due to their various advantages, and their superiority compared to a single method approach. Mixed methods approaches can provide " ... the ability to leverage the strengths of varied methods, provide richer insights into phenomena of interest that cannot be fully understood using only quantitative or qualitative methods, address research questions that call for real-life contextual understanding, multi-level perspectives, and cultural influences" (p. 556). Similarly, [21] highlights the efficiency of mixed method designs in generating holistic understandings of phenomena and for developing novel theoretical clarifications. Of the various purposes underpinning mixed methods designs proposed by [21], this study subscribes to the "developmental purpose". In this purpose, a qualitative study is conducted first with the aim of identifying or developing hypotheses and constructs. A quantitative study is then used to test and validate the hypotheses. In this study, the exploratory study (the qualitative aspect) employs interviews with App purchasers to identify and determine the potential factors that influence purchase decisions. The identified factors are then used to develop a model that explains consumers' decision making when it comes to App purchasing. The confirmatory study (the quantitative aspect) then employs a survey questionnaire to test and validate the proposed model. This research contributes to mobile App research by advancing the understanding of decision factors with regard to mobile App purchases and identifies the interrelationships among these factors by employing a mixed methods approach. The results of this study can subsequently guide the efforts of developers and publishers as to how to generate revenue by stimulating Apps' sales. Mobile App publishers and developers need to identify the hidden factors that drive consumers' decision making when it comes to mobile App purchases. By revealing these key factors, the efforts of both developers and publishers can be directed toward such factors in order to address them. Accordingly, the findings of this study provide important practical implications that can be used to design and develop attractive mobile Apps that improve consumers experience and meet their expectations. Furthermore, the findings of this study may act as a benchmark for other research works that aim to investigate the key factors driving consumers' intention to purchase mobile Apps.

## 3. Exploratory Study

### 3.1. Research Method

The current research follows the generic procedures suggested by Venkatesh et al. [21] to develop mixed methods research. Accordingly, this study firstly adopts interviewees, as a qualitative research method, with the aim of discovering factors that may potentially influence purchasing decisions with regard to paid mobile Apps. As has been highlighted by [22], as a method for collecting data, interviews has various advantages, such as enabling researchers to effectively concentrate on those topics that interest them, and to deliver better explanations and perceptions of causal inferences. In this research, snowball sampling is employed to identify smartphone users who had actually purchased no less than one mobile App in the previous month. While such a criterion directly serves the research objective, it also ensures that the purchase has been performed recently, which therefore prevents the problem of retrospective recall that might occur on the part of the interviewees [23]. Accordingly, a total of 15 interviews (n = 15) were conducted. The sample consisted of 9 professionals (in employment) and 6 students (4 undergraduates, and 2 graduates). The average age of the interviewees was 28.6 years. Additionally, 66.7% (n = 10) of the interviewees were male, and 34.3% (n = 5) were female. All of the interviewees were Apple (n = 9, 60%) and Samsung (n = 6, 40%) phone users. The interviews were conducted on a one-to-one basis in an informal setting, and the average length of the interviews was 25 min. During the interviews, a series of questions was asked with regard to the mobile Apps

she/he had purchased in the previous month and the key factors that guided their decision to purchase. In addition, further questions were asked to explain these factors.

All interviews were recorded and transcribed. The guidelines laid down by Corbin and Strauss [24] in terms of open, axial and selective coding were used to analyze the interview transcripts. The process of coding was conducted by two researchers (coders). During this process, one of the researchers did not engage in data collection, with the aim of preventing any potential bias during the coding process. During the process of open coding, each interview transcript was carefully examined, line by line, by each coder, in order to identify concepts in the textual data that might help explain paid mobile App purchase. Afterwards, both coders intensively and critically discussed all of the identified concepts, which they then labeled (named) after reaching a consensus. While the focus of open coding is to identify emergent concepts, the aim of axial coding is to further sift, align, and refine these emergent concepts in order to categorize them into "distinct thematic categories" [25]. Thus, the axial coding was used to group, relate and classify the concepts (Concept in Table 1) that were identified and elaborated during open coding into broader categories [26]. These broader categories (Category-Level I in Table 1) represent commonalities and similarities among the codes. As a consequence, the identified concepts were integrated into higher-level categories, which therefore resulted in evolving them into theoretical constructs. These theoretical constructs were used to evolve causal relationships that can help explain mobile App purchase.

Finally, selective coding was employed to identify the core categories that can explain the research problem or phenomena and exclude insignificant themes from the final analysis. Specifically, selective coding is concerned with integrating the categories (*Category-Level I* in Table 1) that have been mutually related and elaborated during axial coding, into cohesive core categories (Category-Level II in Table 1) [26]. The significance of selective coding arises from its critical role in enabling researchers to saturate the identified or "selected" categories, while preventing the addition of irrelevant material to the core investigation [27]. It is noteworthy that, in accordance with Guest et al.'s [28] guidelines, data saturation was reached within the first 11 interviews. Practically, information obtained from the interviewees after the 11th interview made very little change to the findings.

## 3.2. Data Analysis and Findings

The open coding process carried out on the transcripts of the interviews has resulted in the identification of a total of 28 concepts (see Table 1). In axial coding, the concepts that emerged from the open coding were classified into broader categories (Category-Level I) and were therefore developed into theoretical constructs. As a matter of fact, axial coding is used to create relationships between core categories and concepts, giving a precise focus on the nature of these relationships (e.g., causal, association). Consequently, eight broad categories were identified in the form of App usefulness, App performance, word-of-mouth (WOM) about the App, price value of the App, App technical reliability, App enjoyment, App trialability together with some others that were later excluded. During selective coding, the core categories that can explain the research phenomenon were identified, and trivial categories were excluded from further analysis. The three core categories that emerged from the selective coding were App-related factors, customer-related factors and marketing-related practices. The core category and the main outcome of interest is mobile App purchase. Therefore, identifying "mobile App purchase" as the core category or "the dependent variable" facilitates the process of relating the seven identified decision factors with the core category.

**Table 1.** Results of open and axial coding.

| Open Coding (Concept) | Axial Coding (Category-Level I) | Selective Coding (Category-Level II) | Frequency * | Percentage |
|---|---|---|---|---|
| Pleasure | App enjoyment | Customer-related factors | 33 | 9.2% |
| Curiosity | | | | |
| Playfulness | | | | |
| Passing time | | | | |
| Reasonable price | Price value of App | | 64 | 17.8% |
| Value for price | | | | |
| Trial version | App trialability | Marketing-related practices | 45 | 12.5% |
| Testing | | | | |
| App reputation | Positive electronic word-of-mouth (eWOM) about App | | 89 | 25% |
| Users' recommendations | | | | |
| App rating | | | | |
| App review | | | | |
| Friends' feedback | | | | |
| Service availability | App technical reliability | App-related factors | 43 | 12% |
| Accurate operation | | | | |
| Service reliability | | | | |
| Utility | App usefulness | | 37 | 10.3% |
| Convenience | | | | |
| Learning | | | | |
| Work | | | | |
| Communication | | | | |
| Functional quality | App performance | | 40 | 11.2% |
| Resource requirements | | | | |
| Functional quality | | | | |
| Processing speed | | | | |
| Personal motivation | Other | - | 7 | 2% |
| Easiness | | | | |
| Self-efficacy | | | | |
| Total | | | 358 | 100% |

* Frequency represents the number of appearances of corresponding concepts in the textual data.

In order to assess the level of agreement between the coders, Cohen's Kappa coefficient and interrater agreement statistics were used [29]. The results show that the interrater agreement was 0.833, indicating that the percentage of agreement between each coder was 83.3%, while the Cohen's Kappa coefficient was 0.824, indicating a substantial agreement between each coder.

With respect to marketing-related practices, the interviewees strongly agreed that positive electronic word-of-mouth (eWOM) was a critical category regarding the decision to purchase paid mobile Apps. Positive eWOM about a mobile App reflects the assessment or evaluation of other users who had experience with the App under consideration. In this category, interviewees indicated that the purchase decision with regard to paid mobile Apps is largely decided by feedback about the Apps

obtained from various information sources. The interviewees highlighted that the purchased Apps were "recommended by friends"; the Apps were "highly rated in the Google Play and Apple Stores"; the Apps were "popular". Furthermore, they pointed out that their purchase decision was significantly driven by taking into consideration "customer reviews", "feedback from other users", and "App rating information in Google Play or App Store". App trialability demonstrates the degree to which the trial/test version of a paid mobile App enables potential users to effectively explore and try the mobile App before purchasing it. With regard to this category, interviewees stated that their purchase decision with regard to paid mobile Apps was motivated by trialability-related aspects. They indicated that the "trial version was a decisive factor that encouraged me to buy the App"; "I loved the App after trying the trial version"; "testing the App allowed me to explore the App and its features before purchasing it".

Regarding customer-related factors, the price value of an App reflects the financial and economic utility resultant from the purchase of a mobile App in comparison to its price. Interviewees in this category stressed that they purchased their mobile Apps because the Apps were "worth their price"; the Apps were "cheap"; "inexpensive"; "affordable"; the Apps were a "bargain"; the Apps had reasonable "value for the price". App enjoyment demonstrates the level of joy and pleasure generated from the use of a particular mobile App. With regard to this category, interviewees declared that they purchased their mobile Apps because they were recognized as being "pleasurable"; "fun"; "playful". Interviewees also state that the Apps triggered their "curiosity" and allowed their "free time" to pass.

With regard to App-related factors, App technical reliability emerged as a critical category. This refers to the degree to which mobile Apps operate adequately without failure. The interviewees stressed that they chose to buy their paid mobile Apps because "service is available 24/7" and they "operate as expected". In addition, they pointed out that the Apps operate without "errors" or "malfunctions". Additionally, App performance emerged as an important category. App performance deals with the overall functionality and quality of the mobile Apps. Interviewees asserted that the decision to purchase their paid mobile Apps was because the Apps did not consume "much of my mobile data"; "storage space on the device". Furthermore, they explained that the Apps operated "fast" with "adequate processing speed". Interviewees also pointed out that the quality of the Apps' functions were "consistent"; "suitable quality"; "with acceptable performance". Another important category that emerged and was regarded as important by interviewees was App usefulness. App usefulness entails the instrumental utility that stems from a mobile App. The interviewees believed that their decision to purchase a paid mobile App was determined by the degree to which the use of a paid mobile App would be beneficial and would increase the users' performance in a specific setting. Among usefulness-related aspects that have been mentioned by interviewees were "the functions of the App were convenient"; "the App is useful" and "the App is important for my work"; "the App's capability of collecting information is advantageous"; "I gained many benefits from using the App in my learning" and "better communication".

Finally, there was a number of concepts (e.g., easiness, personal motivation) related to the category "other" that were excluded as they were not mentioned much by the interviewees (2% of the overall codes). As can be demonstrated, the decision factors identified in Table 2 solely reflect App-related factors (e.g., performance, technical reliability), customer-related factors (e.g., price value of the App, enjoyment), and marketing-related practices (e.g., trialability, positive eWOM). Next, in the confirmatory study, the seven decision factors identified from this exploratory study are interpreted based upon on the related literature. This literature is used to derive the interrelationship between the dependent variable and the identified factors, leading to the creation of the research model as described in the next section.

**Table 2.** Identified decision factors.

| Interviewee | Identified Decision Factors | | | | | | |
| --- | --- | --- | --- | --- | --- | --- | --- |
| | App-Related Factors | | | Customer-Related Factors | | Marketing-Related Practices | |
| | Performance | Usefulness | Technical Reliability | Enjoyment | Price Value | Trialability | Positive eWOM |
| Interviewee 1 | | x | x | | x | | x |
| Interviewee 2 | x | | | x | x | x | x |
| Interviewee 3 | | | x | | x | | x |
| Interviewee 4 | | x | x | | | x | x |
| Interviewee 5 | x | x | | x | x | | x |
| Interviewee 6 | | x | x | | x | | x |
| Interviewee 7 | x | | x | x | | x | x |
| Interviewee 8 | | x | x | | | x | x |
| Interviewee 9 | | | x | | x | x | x |
| Interviewee 10 | | x | | | x | | x |
| Interviewee 11 | x | | x | x | x | | x |
| Interviewee 12 | | | x | x | x | x | x |
| Interviewee 13 | x | x | x | | x | | x |
| Interviewee 14 | | | | | x | | x |
| Interviewee 15 | x | x | | x | x | x | x |

## 4. Confirmatory Study

### 4.1. Research Model and Its Hypotheses

According to the findings obtained from the exploratory study, the research model (see Figure 1) has been proposed. In this model, the seven factors acknowledged earlier are expected to directly influence consumers' intention to purchase paid mobile Apps. The following sections discuss and justify all the hypotheses relating to the research model.

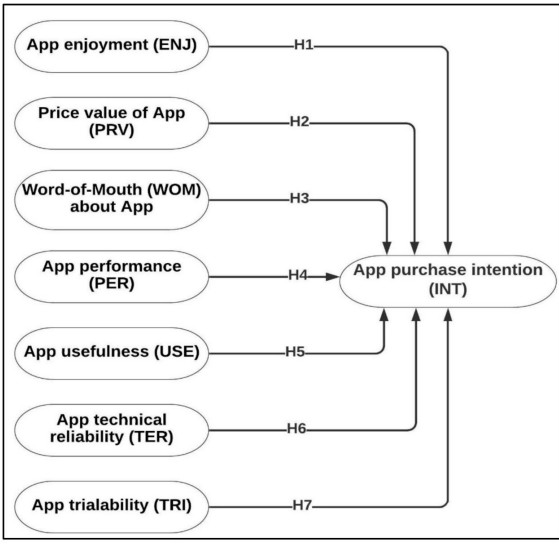

**Figure 1.** The research model.

Davis et al. [30] point out that perceived enjoyment represents " … *the extent to which the activity of using the computer is perceived to be enjoyable in it's own right, apart from any performance consequences that may be anticipated*" (p. 1113). Thus, App enjoyment in this study refers to the sensation of joy, entertainment or pleasure derived from the use of a mobile App. In fact, enjoyment reflects the hedonic (intrinsic) value of a product or service, where the utility stems from affective or feeling states that

the product or service generates [31,32]. Consumers are more likely to purchase products/services depending on feelings of joy and pleasure, mainly in the case of hedonic products (e.g., a mobile game) [33]. Therefore, hedonic value has been recognized as one of the key antecedents that significantly influences how consumers make their purchasing decisions [34]. It has been argued by Tang et al. [11] that "*App consumers focus more on satisfying their own inner spiritual needs rather than targeting practical purpose*" (p. 1632). Further, enjoyment positively influences mobile App usage intentions because, when consumers' entertainment requirements are met, the pleasurable feelings about the App are realized. Hence, consumers may intend to assess a mobile App before purchasing it in anticipation of the level of fun generated from using the App, and its ability to provide pleasure. Thus, the intention to purchase a mobile App is more likely to be increased directly by the mobile App's perceived enjoyment.

**Hypothesis 1 (H1).** *Intention to purchase a mobile App is positively influenced by App enjoyment.*

Xu et al. [35] define price as " . . . *the financial cost required to obtain and use a product*". With regard to mobile Apps, price value has been described by Venkatesh et al. [36] as that price value that serves as " . . . *consumers' cognitive trade-offs between the perceived benefits and cost of using various applications*" (p. 181). Price value is perceived by consumers on the basis of their beliefs of what is received in return for what is given (monetary investment). Hence, the increase in consumer returns as compared to consumer investment will therefore lead to a perception of increased monetary value. It has been found that price value is among the core factors that determines mobile App purchase [32,37,38], especially in light of the fact that the current market for mobile Apps is dominated by free Apps. In mobile App stores, there are several different Apps with comparable functions, and the majority of these Apps are free, which subsequently minimizes consumers' motivation to purchase paid mobile Apps with the same functions [11]. This assumption holds true even if paid mobile Apps provide an improved functional quality in comparison with other Apps. Consumers are inclined to assess the overall expected price against the overall expected gain of the paid mobile Apps. When an App's performance surpasses its price, the most likely result is for it to be purchased. Otherwise, it will suffer from poor adoption rates. Consumers who consider purchasing a mobile App are more likely to evaluate its value for its price and purchase it if its value is high. Therefore, price value of paid mobile Apps is assumed to have direct role in increasing consumers' intention to purchase such Apps.

**Hypothesis 2 (H2).** *Intention to purchase a mobile App is positively influenced by App price value.*

Huete-Alcocer [39] states that word of mouth (WOM) is referred to as "*communication between consumers about a product, service, or company in which the sources are considered independent of commercial influence*" (p. 1). Such interpersonal and informal exchanges offer access to vital information regarding the consumption of products/services in addition to formal advertising. In other words, WOM adds a layer to the messages offered by sellers, and influences the decision making of consumers. WOM is broadly viewed as one of the most critical factors driving consumer behavior [40]. As a result, WOM is recognized by various researchers [41,42] as the most significant source of information in terms of consumer buying decisions. Abubakar and Ilkan [43] point out that the emergence of online platforms has resulted in extending WOM to electronic WOM (eWOM), which is now regarded as one of the most significant information sources on the Web. In the case of eWOM, information is offered by consumers through the Internet about products/services [44]. Specifically, consumers post and share their recommendations, comments, reviews and suggestions about products/services on the internet through various tools such as e-bulletin systems, review sites, social networking sites, discussion platforms and forums. WOM communication can have both positive (PWOM) and negative impacts (NWOM). Al-Adwan and Kokash [45] point out that PWOM " . . . *is triggered when there are social cues that deliver positive signs such as significant number of customers who have purchased or intended to purchase a product or service*" (p. 20). PWOM is recognized by marketing experts as an effective marketing tool as it generates positive opinions and feedback that can affect consumers' decisions to

purchase the brand in question [46]. On the other hand, opposite effects can be generated by NWOM. The conceptualization of WOM with regard to mobile Apps in this study is described as the electronic or traditional communication of positive assessment provided by other consumers about the App in question. The content of most eWOM is generated by genuine customers who are independent of the market [39]. Therefore, this independency makes eWOM about an App a more credible and reliable medium by decreasing risk and uncertainty about the App with respect to purchasing and using it. As a consequence, this practice of reassurance and guarantee could result in the development of consumers' trust in the App under consideration [2]. Accordingly, eWOM about an App can significantly influence its reference price in and raise the value of the deal. Based on such a perception of value, consumers are inclined to purchase the App in question. Hence, positive eWOM about an App is expected to improve consumers' intention to purchase that paid mobile App.

**Hypothesis 3 (H3).** *Intention to purchase a mobile App is positively influenced by positive eWOM about an App.*

Performance is recognized as a critical aspect in measuring mobile Apps' service quality [47]. The theory of expectancy-confirmation [48] has been developed to explain consumer satisfaction. It is suggested that consumers develop expectations with regard to the performance of products/services before purchasing and will make their mind up about perceived performance after obtaining the products/services [49]. Consumers will then compare their expectations and perceived performance. They will be satisfied if the perceived performance exceeds their expectations. On the other hand, if expectations are less than the perceived performance, a negative disconfirmation will be generated, and therefore consumers will be less satisfied. Similar to [32], this study considers App performance in terms of the extent to which a mobile App is believed to have functional value depending on performance expectation and perceived quality. A high perception of performance is achieved when the performance of a mobile App surpasses its expected performance. Furthermore, mobile Apps are expected to operate smoothly without dropout or malfunction. Wulfert [47] states that the performance of mobile Apps is weighted by the functional quality (the processing speed) of the App, and its resource requirements. In terms of resource requirements, two main aspects are identified: mobile network usage and device storage. The processing speed of mobile Apps relates to the processing performance of any operation or function with regard to mobile Apps, including instant page loading and transitions, smooth scrolling and quick responses to the customer's inputs [50]. Moreover, Madu and Madu [51] point out that processing speed is related to the ability to download information. It also relates to the quality of data processing and transfer [52]. Device storage usage is concerned with the use of disk space on the mobile device resulting from the downloading of mobile Apps. High memory usage is recognized as a key issue when deciding whether or not to download a particular mobile App [53]. Given that the storage capacity of mobile devices is limited [54], and most devices cannot be extended in terms of extra memory capacity, mobile Apps should have an appropriate size. Such a size indicates that mobile Apps should take up as little disk space as possible when it comes to providing their mobile services (m-services) to consumers [47]. The usage of mobile network data relates to the traffic that mobile Apps cause [55]. It measures the network traffic it causes for the mobile cellular network, and the consumers' volume of purchased data. It is critical that mobile Apps cause as little mobile network traffic as possible, as well as the traffic required to provide their information and features. Therefore, App performance is expected to increase consumers' intention to purchase paid mobile Apps.

**Hypothesis 4 (H4).** *Intention to purchase a mobile App is positively influenced by App performance.*

Recent research indicates that perceived usefulness has an essential role on users' intention to adopt mobile Apps. Based on previous research [56], App usefulness refers to what extent the use of a mobile App improves performance in accomplishing what she/he wishes to do. Kim et al. [2]

define perceived usefulness as " . . . *the functional value of a good, which is the utility derived from the perceived quality and expected performance of the good*". Functional value reflects the level to which a product's features satisfy the utilitarian needs of consumers [57]. It has been reported that functional value is recognized as a major decisive factor regarding consumers' purchasing decisions [37,58,59]. When considering the purchase of a mobile App, individuals are inclined to evaluate whether or not the App is beneficial and useful in terms of what they want to achieve. Consequently, consumers' intentions to purchase a mobile App are expected to be directly increased by the App's perceived usefulness.

**Hypothesis 5 (H5).** *Intention to purchase a mobile App is positively influenced by App usefulness.*

Technical reliability represents the technical features of the reliability of mobile Apps [47]. It focuses on ensuring the consistency and accuracy of mobile Apps' operations and services. Mobile Apps are expected to operate on mobile devices without any failure or dropout. According to the Institute of Electrical and Electronics Engineers (IEEE) [60], reliability is described as " . . . *the ability of a system or component to perform its required functions under stated conditions for a specific period of time*". However, technical reliability is referred to by Parasuraman et al. [61] as "system availability". Al-Kuwaiti et al. [62] point out that reliability refers " . . . *to failure-free operation during an interval, availability refers to failure-free operation at a given instant time*" (p. 113). Accordingly, one can argue that availability can be considered as an evaluation of reliability at a particular point in time. Wulfert [47] points out that technical reliability can be broken down into two key sub-dimensions: the m-service availability and the feature reliability of mobile Apps. Feature reliability is concerned with measuring the promised performance and adequacy of operation of a mobile App. It particularly highlights the mobile App's reliability in terms of constant operation without any malfunctions, and the appropriate start of the mobile App. Availability refers to the percentage of time during which the system is fully operational and available. It is important that the m-services offered through the mobile App must be available at any time consumers need to use and access them. Furthermore, since there are no spatial and temporal constraints in terms of accessing m-services, such services have to be available. As m-services can be accessed without any temporal and spatial constraints, the availability of such service is deemed to be necessary any time. It has been suggested that the unavailability of m-services adversely impacts consumers' perceptions with regard to mobile Apps' technical reliability.

**Hypothesis 6 (H6).** *Intention to purchase a mobile App is positively influenced by App technical reliability.*

According to the theory of innovation diffusion [63], trialability indicates that individuals need to personally experience and try out an innovation in such a way as to entirely test that innovation before adopting it. Having the opportunity to try an innovation enables users to endorse expectations and develop beliefs as to the extent to which the innovation can meet their personal needs [64]. Thus, it allows users to proactively avoid any potential costs and commitment by trying innovations prior to adopting them. This study describes mobile App trialability as the extent to which potential consumers believes that they can sufficiently test and try mobile Apps prior to purchase [2]. A free trial of a new technology or system (such as mobile Apps) is effective in diminishing uncertainty and ambiguity, allowing potential users to arrive at a purchasing decision [65]. Trial versions allow users to test and examine products and resolve any uncertainty about the products' real value, such as ease of use and usefulness [1]. Furthermore, trialability can act as an indication of product quality; knowing that products are available in free trial versions may act as a guarantee to consumers. Since consumers place higher weight on certain outcomes rather than merely probable ones, trialability increases the outcome certainty, and subsequently influences consumers' intention to purchase paid mobile Apps.

**Hypothesis 7 (H7).** *Intention to purchase a mobile App is positively influenced by App trialability.*

### 4.2. Research Method

To empirically validate the hypotheses of the research model, an online survey questionnaire was used. The questionnaire was hosted by Survey Monkey (an online survey provider). The direct link of the survey was distributed to the participants in this study who are based in Jordan, via two channels in the form of e-mail and social media networking sites (e.g., Facebook, Instagram). Convenience sampling was used to select the participants. Cohen et al. [66] state that convenience sampling is recognized as an economic and fast method of sampling that enables researchers to easily access available participants. Such a sampling method has been criticized due to its inability to produce a representative sample. However, this method of sampling is deemed to be suitable for exploratory studies such as the current one, where there is no emphasis on generating any sort of generalization. To serve the objective of this study, the targeted participants were smartphone users. Accordingly, at the beginning of the survey, a screening question was included to decide whether or not the participants satisfied the inclusion condition. The link of the survey was posted and emailed to a total of 700 participants. In total, 511 responded and returned the questionnaire, giving a response rate of 73%. After screening the returned responses, a total of 27 responses were eliminated due to incompleteness (80% of data in these responses were reported as missing). As a result, a total of 486 responses were valid and qualified for progress to the analysis stage. According to Hair et al. [67], this number of responses satisfies the rule of thumb for the minimum sample size required for conducting structure equation modeling (SEM).

The survey questionnaire was divided into two main parts. The first part was devoted to enquiring about the participants' demographics (see Table 3).

**Table 3.** Descriptive statistics of participants.

| Demographic Measure | | Frequency | % |
|---|---|---|---|
| Gender | Male | 275 | 56.6 |
| | Female | 211 | 54.4 |
| Age | <20 | 33 | 6.7 |
| | 21–25 | 194 | 40 |
| | 26–30 | 168 | 34.6 |
| | >30 | 91 | 18.7 |
| Platform | iPhone OS | 258 | 53 |
| | Android | 194 | 40 |
| | Windows phone | 34 | 7 |
| | Other | 0 | 0 |
| Usage experience | <1 year | 18 | 3.7 |
| | 1–3 years | 159 | 32.7 |
| | >3 years | 309 | 63.6 |
| Profession | Students | 196 | 4.3 |
| | Professional | 263 | 54.1 |
| | Not employed | 27 | 5.6 |

*N* = 486.

In the second part, sets of feature questions (measurement items) were developed to measure the research model's factors (constructs). The development of the measurement items (Table 4) was informed by the related literature. Specifically, the measurement items were developed by adapting validated instruments from the existing literature. The original items were adjusted and refined to match this study's context and objective. In order to measure each factor's items, a five-point Likert scale ranging from "1 = strongly disagree" to "5 = strongly disagree" was employed. The dependent variable of the research model was the intention to purchase a mobile App, and thus the participants were requested to think of a specific mobile App at the beginning of the survey. The selection of the intention to purchase as the dependent variable was driven by the fact that information about the

behavior of actual purchase was unavailable. Previous scholars have argued that behavioral intention is regarded as the utmost important aspect of a particular behavior [68]. Two procedures were followed to ensure the appropriateness and validity of the measurement items. First, the measurement items were introduced and evaluated by an academic panel consisting of four individuals who have expertise in the field. The evaluation process indicated that the level of agreement among the members of the panel was 87.5%. Furthermore, various suggestions offered by the panel were considered with the aim of improving the readability and reliability of the items. Second, a pilot study was conducted before the submission of the final survey questionnaire to the participants. A total of 50 respondents involved in the pilot study were used to evaluate the measurement items with respect to reliability and readability. The results demonstrated that the respondents were of the opinion that the items were reliable and clear. In addition, all factors had an acceptable internal consistency as each factor had a Cronbach Alpha < 0.7 [67].

**Table 4.** Measurement Items.

| Factor | Item | Source |
|---|---|---|
| App usefulness (USE) | USE1: "I think using the mobile App enables me to accomplish tasks more quickly". USE2: "I think the mobile App is useful in my daily life". USE3: "Using the App saves me time and effort in getting done what I want". USE4: "Using the App makes it easier to do what I want". | [2,69] |
| App enjoyment (ENJ) | ENJ1: "I think using the mobile App is entertaining". ENJ2: "I think using the mobile App is fun for me". ENJ3: "I have fun interacting with the mobile App". | [1,2] |
| App trialability (TRI) | TRI1: "Before deciding on whether or not to purchase the mobile App, I am able to try it out properly". TRI2: "I am permitted to use the mobile application on the mobile phone on a trial basis long enough to see what it could do". TRI3: "I am permitted to use the mobile App on a trial basis long enough to see what it could do". TRI4: "I have a great deal of opportunity to try the App". | [1,64] |
| App performance (PER) | PER1: "I think the mobile App reacts responsively to my interactions". PER2: "I think the mobile App occupies as little storage as possible". PER3: "I think the mobile App causes as little network traffic as possible". PER4: "I think the mobile App offers consistent quality". | [32,47] |
| Price value of App (PRV) | PRV1: "The App is reasonably priced". PRV2: "The App is good value for money". PRV3: "At the price shown, the App is economical". | [2,70] |
| App technical reliability (TER) | TER1: "I believe it is important that the mobile App will start up quickly without a long wait". TER2: "I think it is important that the m-services provided will be available anytime I want to access them". TER 3: "I think it is important that the mobile App will be executed according to the given description and promises". TER4: "I think it is important that the mobile App will operate properly after updating". | [47] |
| Electronic Word-of-Mouth about App (eWOM) | eWOM1: "Many users say good things about the mobile App". eWOM2: "The mobile App is highly rated". eWOM3: "Many users recommend the use of the mobile App". eWOM4: "Many users provided positive reviews about the mobile App". | [2,32] |
| App purchase intention (INT) | INT1: "I am very willing to purchase the mobile App in the near future". INT2: "I find purchasing the mobile App to be worthwhile". INT3: "There is a high probability that I will consider purchasing the mobile App in the future". | [2,11] |

## 4.3. Empirical Results

### 4.3.1. Preliminary Analysis

Prior to proceeding to the data analysis, many item analysis and data screening issues were addressed by using SPSS (version 23). These issues include internal consistency reliability, normality and multi-collinearity. The reliability in terms of internal consistency was weighed by the use of Cronbach Alpha ($\alpha$). Normality was determined by evaluating the skewness and kurtosis estimates for each

item. Finally, multi-collinearity was weighted by inspecting the value of the "Variance Inflation Factor" (VIF). According to [67], the minimum acceptable cutoff point for $\alpha$ is 0.7, and the VIF acceptable point should be <3. Furthermore, the acceptable values of skewness and kurtosis should be between $-2$ and $+2$ [71]. As is demonstrated in Table 5, the values of $\alpha$, VIF, skewness and kurtosis were all within the recommended values. It can therefore be concluded that each construct has adequate internal consistency, and no serious problems in terms of normality were present. As this study is based on a single source of data, common method bias (CMB) was a potential issue. Thus, following the recommendation of Podsakoff et al. [72], Harman's one factor test was conducted to ensure the absence of CMB. An exploratory factor analysis was performed, and all items were factorized into a single factor. The results indicate that eight factors emerged, as none of these factors accounts for $\geq$50% of variance among the measurement items. Accordingly, the absence of CMB was evident.

**Table 5.** Preliminary analysis.

| Construct | Item | Mean | Std. | $\alpha$ | VIF | Skewness | Kurtosis |
|---|---|---|---|---|---|---|---|
| App performance (PER) | PER1 | 1.7 | 0.57 | 0.92 | 2.81 | 0.77 | 1.35 |
| | PER2 | | | | | 0.65 | 1.92 |
| | PER3 | | | | | 0.73 | 1.64 |
| | PER4 | | | | | 0.72 | 1.58 |
| App usefulness (USE) | USE1 | 1.6 | 0.6 | 0.94 | 2.88 | 1.01 | 1.97 |
| | USE2 | | | | | 0.96 | 1.94 |
| | USE3 | | | | | 1.12 | 1.96 |
| | USE4 | | | | | 0.90 | 1.98 |
| App enjoyment (ENJ) | ENJ1 | 1.66 | 0.57 | 0.93 | 2.33 | 0.68 | 1.82 |
| | ENJ3 | | | | | 0.64 | 1.81 |
| | ENJ2 | | | | | 0.73 | 1.78 |
| App technical reliability (TER) | TER2 | 1.63 | 0.54 | 0.90 | 2.6 | 0.84 | 1.65 |
| | TER3 | | | | | 0.77 | 1.64 |
| | TER1 | | | | | 0.82 | 1.69 |
| | TER4 | | | | | 0.73 | 1.72 |
| App purchase intention (INT) | INT3 | 1.65 | 0.59 | 0.91 | – | 0.89 | 1.62 |
| | INT1 | | | | | 0.84 | 1.56 |
| | INT2 | | | | | 0.81 | 1.56 |
| App trialability (TRI) | TRI1 | 1.64 | 0.58 | 0.93 | 2.57 | 0.86 | 1.81 |
| | TRI2 | | | | | 0.87 | 1.68 |
| | TRI3 | | | | | 0.91 | 1.57 |
| | TRI4 | | | | | 0.80 | 1.70 |
| Price value of App (PRV) | PRV1 | 1.66 | 0.61 | 0.93 | 2.68 | 0.76 | 1.09 |
| | PRV2 | | | | | 0.81 | 1.33 |
| | PRV3 | | | | | 0.73 | 1.26 |
| Electronic word-of-mouth about App (eWOM) | eWOM1 | 1.67 | 0.58 | 0.92 | 2.66 | 0.80 | 1.45 |
| | eWOM2 | | | | | 0.83 | 1.98 |
| | eWOM3 | | | | | 0.77 | 1.30 |
| | eWOM4 | | | | | 0.89 | 1.61 |

Std. "standard deviation", $\alpha$: "Cronbach Alpha", VIF: "variance inflation factor".

SEM—"Structural Equation Modeling" technique was applied to analyze the proposed model as shown in Figure 1. Specifically, Partial Least Square-SEM (PLS-SEM) was the selected approach as it is suitable for use in exploratory research and theory development, and because it is appropriate for non-normal data [73]. Thus, Smart-PLS software version 3.2.9 [74] was employed to examine the research model. Corresponding to Henseler et al. [75], the PLS approach includes a two-step procedure that includes the assessment of the: (1) the outer measurement model, and (2) the inner structural model.

### 4.3.2. Measurement Model

In this stage, confirmatory factor analysis (CFA) was conducted to determine whether or not the research model's construct has acceptable validity and reliability measures. In terms of construct validity, two examinations were performed: convergent validity and discriminant validity. Convergent validity was examined by the means of AVE—"average variance extracted" and item loadings. Hair et al. [67] suggest that the minimum threshold of AVE is 0.5, and each item's loading should load on its intended construct with a value higher than 0.7. Table 6 shows the cross-loadings and confirms that the items loaded substantially (>0.7) on their intended construct. In addition, the minimum acceptable value of AVE was exceeded by all the constructs. Hence, discriminant validity was confirmed. Furthermore, composite reliability (as another weight of internal consistency) was examined. As can be shown in Table 6, the composite reliability (CR) of all the constructs surpassed the acceptable coefficient of 0.7.

**Table 6.** Construct reliability and validity.

| Construct | Item | Loading | CR | AVE |
|---|---|---|---|---|
| App enjoyment (ENJ) | ENJ1 | 0.93 | 0.96 | 0.88 |
| | ENJ2 | 0.95 | | |
| | ENJ3 | 0.94 | | |
| App purchase intention (INT) | INT1 | 0.92 | 0.94 | 0.85 |
| | INT2 | 0.91 | | |
| | INT3 | 0.93 | | |
| App performance (PER) | PER1 | 0.92 | 0.95 | 0.82 |
| | PER2 | 0.89 | | |
| | PER3 | 0.90 | | |
| | PER4 | 0.91 | | |
| Price value of App (PRV) | PRV1 | 0.95 | 0.96 | 0.90 |
| | PRV2 | 0.94 | | |
| | PRV3 | 0.93 | | |
| App technical reliability (TER) | TER1 | 0.87 | 0.93 | 0.77 |
| | TER2 | 0.89 | | |
| | TER3 | 0.88 | | |
| | TER4 | 0.86 | | |
| App trialability (TRI) | TRI1 | 0.92 | 0.95 | 0.84 |
| | TRI2 | 0.92 | | |
| | TRI3 | 0.91 | | |
| | TRI4 | 0.89 | | |
| App usefulness (USE) | USE1 | 0.91 | 0.96 | 0.85 |
| | USE2 | 0.92 | | |
| | USE3 | 0.93 | | |
| | USE4 | 0.92 | | |
| Electronic word-of-mouth about App (eWOM) | eWOM1 | 0.91 | 0.94 | 0.81 |
| | eWOM2 | 0.88 | | |
| | eWOM3 | 0.90 | | |
| | eWOM4 | 0.89 | | |

CR: "composite reliability", AVE: "Average variance explained".

Discriminant validity was checked by means of Fornell and Larcker's criteria [76]. These criteria suggest that the value of each construct's $\sqrt{AVE}$ should exceed the correlation coefficient of any other construct in the model. The results shown in Table 7 confirm that such a condition was satisfied, which therefore indicates that discriminant validity was present.

**Table 7.** Discriminant validity.

| Construct | ENJ | INT | PER | PRV | TER | TRI | USA | eWOM |
|:---:|:---:|:---:|:---:|:---:|:---:|:---:|:---:|:---:|
| ENJ | * 0.942 | | | | | | | |
| INT | ** 0.738 | 0.93 | | | | | | |
| PER | 0.66 | 0.78 | 0.91 | | | | | |
| PRV | 0.62 | 0.80 | 0.68 | 0.94 | | | | |
| TER | 0.66 | 0.75 | 0.69 | 0.63 | 0.88 | | | |
| TRI | 0.62 | 0.79 | 0.68 | 0.68 | 0.64 | 0.92 | | |
| USA | 0.65 | 0.79 | 0.69 | 0.70 | 0.67 | 0.70 | 0.92 | |
| eWOM | 0.65 | 0.77 | 0.68 | 0.69 | 0.69 | 0.64 | 0.67 | 0.90 |

* "The numbers on the leading diagonal are the square root of AVE for each construct". ** "Correlation among constructs".

### 4.3.3. Structural Model

Before examining the proposed model, fit indices were assessed by the means of three main model fit measures: the SRMR—"Standardized Root Mean Square Residual", the NFI—"Normed Fit Index", and the exact model fit. While the SRMR represents a goodness of fit weight that can be utilized to overcome the issue of model misspecification [77], NFI measures the model's incremental fit [67]. As shown in Table 8, the value of SRMR for this study was less than 0.05, while the NFI value was greater than 0.9, which indicate a good fit [78]. Finally, the exact model fit evaluates the discrepancy between the covariance matrices implied by the empirical data, and the composite factor model. Dijkstra and Henseler [79] have suggested two different ways to calculate such a discrepancy: d_LS, "the squared Euclidean distance", and d_G "the geodesic distance". This test (see Table 8) indicates that "dULS < bootstrapped HI 95% of dULS and dG < bootstrapped HI 95% of dG" [78], demonstrating suitable fitness between the model and the data.

**Table 8.** Model fit.

| Fit Index | Value |
|:---:|:---:|
| SRMR | 0.023 |
| dULS | 0.24 |
| dG | 0.359 |
| NFI | 0.942 |

After assessing the model fit, the estimation of the research model was interpreted by applying SEM. As Figure 2 and Table 9 indicate, the seven hypotheses were supported by the data. Specifically, each of enjoyment, price value, word of mouth, performance, usefulness, technical reliability and trialability had significant positive effects on App purchase intention. Altogether, these constructs explained 84.4% ($R^2 = 0.844$) of the variance in INT, demonstrating a substernal explanation [67] for consumers' intention to purchase paid Apps.

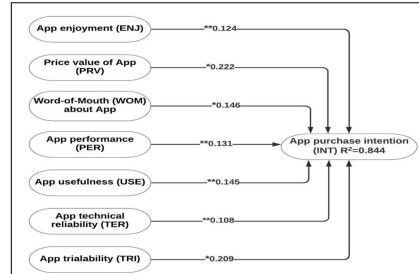

**Figure 2.** Path analysis, * *p*-value < 0.000, ** *p*-value < 0.01.

**Table 9.** Structural model analysis.

| Hypothesis | Path | β | *T*-Value | *p*-Values | Result |
|:---:|:---:|:---:|:---:|:---:|:---:|
| H1 | ENJ -> INT | 0.124 | 2.708 | 0.007 | Supported |
| H2 | PRV -> INT | 0.222 | 4.824 | 0.000 | Supported |
| H3 | eWOM -> INT | 0.146 | 3.743 | 0.000 | Supported |
| H4 | PER -> INT | 0.13 | 3.166 | 0.002 | Supported |
| H5 | USA -> INT | 0.145 | 2.856 | 0.004 | Supported |
| H6 | TER -> INT | 0.108 | 3.039 | 0.002 | Supported |
| H7 | TRI -> INT | 0.209 | 4.527 | 0.000 | Supported |

While price value had the highest effect on purchase intention (β = 0.222, *p*-value < 0.000), technical reliability had the lowest effect (β = 0.108, *p*-value < 0.01). Table 9 displays each path coefficient (hypothesis) and its corresponding *t*-value. These results confirm the findings of the exploratory study detailed in Section 3.2.

## 5. Discussion

Both App enjoyment and the price value of the App represent customer-related factors. Consistent with earlier research [2,32], the results reveal that the price value of an App is a decisive factor for App purchasing intentions. This finding is also in line with previous research which confirms the significant role of price value in boosting the behavioral intentions with regard to using mobile Apps [80]. It has a positive relationship with purchasing intentions. This means that the more consumers believe that Apps are reasonably priced compared to their benefits and values, the more they have favorable intentions to purchase them. Consumers are generally reluctant to purchase highly priced mobile Apps, especially in light of the existence of free alternatives with similar functionalities. Thus, consumers tend to assess the overall price against the overall expected benefits/values of the App under consideration. If an App's benefits/value outweigh its price, the possibility of it being purchased is higher; other than that, it will face poor download and adoption rates. Since a varied set of free Apps has been developed to meet consumers' demands and requirements, only exceptional and high quality paid Apps can lead to consumers' intentions to purchase.

Similar to previous research on mobile Apps [38], App enjoyment is found to be another important factor that influences purchasing intentions. Furthermore, a previous study [11,81] supports the importance of perceived enjoyment with regard to behavioral intentions to download mobile Apps. Enjoyment has a positive relationship with consumers' purchasing intentions. This points to the assumption that the greater the increase in consumers' perception of an App's enjoyment, the more likely is the intention to purchase it. Enjoyment perceptions relate to the hedonic utility perceived by users in terms of feelings of satisfaction and uniqueness. Unlike in the case of physical products, App users are more concerned with fulfilling their "inner spiritual needs" rather than pursuing practical purposes.

In line with [2], it has been found that there is a significant positive relationship between App trialability and purchasing intentions. Many studies have demonstrated the important impact of trialability on behavioral intentions to adopt mobile Apps [1,82]. It is common for potential users to avoid risks and make a conservative choice due to their uncertainties and doubts about the Apps' functionality, especially in the case of highly priced Apps. In fact, having users try out paid Apps before they purchase, plays an imperative role in steering purchase decision making. Before purchasing a mobile App, users form expectations about the performance of the App. Trial versions enable users to have actual experience with the App, and allow them to compare the expected and actual performance of the App. Such free trials have a critical effect in reducing the level of uncertainty and risk, and allowing users to become familiar with App's functionality and content, which therefore helps when it comes to deciding whether or not to purchase the App.

Positive electronic word-of-mouth (eWOM) about Apps is recognized as a critical factor when it comes to purchasing mobile Apps. It has a significant positive effect on purchasing intentions, which is in line with [32] and inconsistent with [2]. This implies that when consumers are exposed to a positive eWOM about a mobile App, they become more willing to purchase that App in the future. Consumers encounter difficulties when it comes to making the right purchase decision regarding technology-related products, such as mobile Apps, due to the lack of experience of the actual features and characteristics of the product [83]. In such a scenario, the seeking of reviews and opinions pre-purchase is critical for potential consumers. Thus, eWOM, as a form of social influence, is a rich and reliable source of product-related information that provides consumers with a valuable source of expert and customer opinion about products. Moreover, eWOM opinions are collected from multiple customers which, as a result, supports richness, truthfulness and impartiality.

Furthermore, the significant positive effect of App usefulness on purchasing intentions is empirically confirmed in this study. This finding is supported by [84], and other studies that validate the importance of usefulness in terms of mobile App adoption [85]. Such a finding indicates that the more a mobile App is perceived by consumers as being useful and having functional value, the greater the intention to purchase. Finally, the technical reliability and performance of mobile Apps are shown to have significant positive effects on purchasing intentions. These findings are supported by those of [47,86]. Such findings indicate that the more a mobile App is perceived as being technically reliable and dependable when it comes to delivering its intended functions and services, the more likely it is that consumers will have favorable intentions to purchase the App. It is important that potential consumers perceive mobile Apps as being available and operating reliably during the App usage cycle.

## 6. Practical and Managerial Implications

The findings suggest several significant practical implications. Price value is seen as a critical factor in determining mobile App purchase intention. Thus, software and mobile App developers should focus their efforts on producing high-availability and multifunctioning mobile Apps, optimizing effectiveness and efficiency while minimizing price. Additionally, they need to highlight comparative advantages, in an effort to have more consumers feeling obliged to adopt paid Apps rather than free ones, which are available at no upfront cost for users [87]. Enjoyment is found as an important factor that influence mobile App purchase intention. Accordingly, it is recommended that developers should attempt to stimulate App consumers' perceived enjoyment, curiosity, pleasure and fun. These components should be reflected in the app's functionality and design. Trialability is introduced as another decisive factor that significantly impacts users' intention toward purchasing mobile Apps. Consequently, App developers and publishers should consider releasing a low-priced or free trial version for a limited time before the paid App (especially highly priced ones) is released. This provides a proper determination as to whether or not to purchase a specific App based on users' needs. By doing so, the purchase burden resulting from users' doubts and high-risk perceptions regarding an App's functionality and performance can be eliminated.

The results highlight eWOM as a fundamental factor in driving mobile App purchase intention. Mobile App publishers and vendors should motivate users to make a positive eWOM (e.g., through reviews, ratings, recommendations) about their Apps. Further, they should pay attention to the fact that positive eWOM has a great impact on potential consumers. Potential consumers consider the various forms of eWOM as trusted sources of information about mobile Apps. Consequently, peer feedback and reviews are deemed to be particularly influential for mobile Apps purchasing decisions. Additionally, negative comments and reviews should be carefully handled, and complaints should be resolved. The significant influence of perceived usefulness is also confirmed. Hence, mobile developers and vendors should bear in mind that App usefulness is a substantial factor in terms of purchasing intentions. App usefulness can be improved in the pre-usage phase by concentrating on forming consumers' beliefs about the quality and functional value of the mobile App. Importantly, consumers should recognize the purpose of the mobile App, and believe that it can deliver its intended functions. To do so, it is necessary to establish and design appropriate communication campaigns and channels that offer consumers adequate information about the aesthetic, quality, functional and technical features of mobile Apps, especially newly released ones. These beliefs can significantly shape and develop the perception of App usefulness. Technical reliability and performance are recognized as vital when it comes to purchasing mobile Apps. Therefore, mobile App developers should focus their efforts on ensuring that mobile Apps should execute their intended m-services instantly and accurately, in conformity with the given descriptions. Moreover, when developers release the new version of their mobile Apps, they have to ensure a firm and consistent quality for each release.

## 7. Theoretical Contribution

Two-fold theoretical contributions are offered by this research. Primarily, this study investigates the decision factors that influence paid mobile App purchases from the consumers' viewpoint and extends the existing literature with regard to mobile Apps by identifying and testing these factors. While a considerable amount of research has been devoted to investigating mobile App adoption and mobile online purchasing, there is a scarcity of studies investigating mobile App purchasing. Importantly, mobile App purchasing is different from other mobile App-related phenomena as it has its unique characteristics. Secondly, unlike much mobile App-related research, the study avoids falling into the trap of picking a list of factors that have already been empirically tested and validated as factors that influence mobile App purchasing intentions, and thereby generates misleading findings. In addition, this study avoids the use of well-known theories employed to investigate mobile App adoption, as such theories may not be particularly relevant. Instead, this study adopts a sequential mixed-methods approach, qualitative followed by quantitative, pursuing systematic, accurate and complete identification and understanding of the decision factors relating to mobile App purchase.

## 8. Conclusions and Future Work

The main objective of this study was to explore what factors consumers consider when it comes to deciding on purchasing mobile Apps. To achieve this objective, a mixed methods research design was adopted. It consisted of an exploratory study (qualitative) followed by a confirmatory study (quantitative). The results that emerged from this research represent various valuable findings for App developers, publishers and marketers in terms of how to stimulate the purchase of their Apps and subsequently attain higher sales. The findings demonstrate that seven decision factors have a direct impact on mobile App purchase intention. These factors include the price value of the App, App performance, App enjoyment, App trialability, electronic word-of-mouth (eWOM) about the App, App technical reliability and App usefulness. This study was conducted to investigate consumers' purchasing intentions. However, the initial purchase intention differs from the actual purchase. There may be different decision factors that influence the actual purchase of paid Apps. Accordingly, additional research is needed to highlight the differences of these two scenarios. Finally, this study made an empirical analysis of the mobile App purchasing intentions of Jordanian consumers,

without bearing in mind differences in research outcomes that might result from regional and cultural differences. Further research could examine and validate the factors identified in this study in various countries to conduct comparative cross-country studies.

**Author Contributions:** Conceptualization, A.S.A.-A. and G.S.; methodology, A.S.A.-A.; validation, A.S.A.-A.; formal analysis, A.S.A.-A.; investigation, A.S.A.-A.; resources, A.S.A.-A. and G.S.; data curation, A.S.A.-A.; writing—original draft preparation, A.S.A.-A.; writing—review and editing, A.S.A.-A.; visualization, A.S.A.-A. and G.S.; supervision, A.S.A.-A.; project administration, A.S.A.-A.; funding acquisition, A.S.A.-A. and G.S. All authors have read and agreed to the published version of the manuscript.

**Funding:** This research was funded by Al Ahliyya Amman University, Jordan.

**Acknowledgments:** We would like to thankfully acknowledge the assistance and support of the faculty members of the departments of Electronic Business & Commerce, and Management Information Systems at Al-Ahliyya Amman University for providing insightful feedback.

**Conflicts of Interest:** The authors declare no conflict of interest.

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
