# Peer review of "What Makes Consumers Purchase Mobile Apps: Evidence from Jordan"

_jtaer, doi:10.3390/jtaer16030034_

Round 1
Reviewer 1 Report
This study uses the mixed-methods approach to analyze the factors that influence consumers’ decisions to purchase mobile Apps. An exploratory study of interviews with 18 consumers is first conducted to determine the factors that influence consumers’ purchase behavior decisions. Then, a confirmatory study that employs a quantitative approach is undertaken to test the proposed model.
However, I have some serious concerns. This requires authors to make a serious revision.
- The abstract should reflect the innovation of this paper, especially innovative conclusions, but the background in the current abstract seems redundant and cannot highlight the innovation and contribution of this paper.
- The current status of mobile apps and paid mobile apps is presented in the Introduction. It is also pointed out that paid mobile Apps and free mobile Apps are different, but the differences are not mentioned. The authors should point out the differences to make this paper meaningful.
- The theoretical contribution of this paper is presented as a separate Section 5, which should be highlighted in the Introduction.
- Discussion and practical implications in Section 4 should be stated separately to make the presentation clearer. Discussion following practical implications seems disorderly. It can be a separate paragraph for practical implications.
- The qualitative study of interview and the quantitative study of questionnaire s are based on Jordanian consumers, so the title, abstract, and keywords need to be revised to add spatial scope.
- As regards the research model and measurement items, I have some
- Line 256, “Price value is perceived by consumers on the basis of their beliefs of what is received in return 256 for what is given (monetary investment)”, price value seems to be a concept about after consumption. The measurement item of price value is according to consumers’ feelings after use the mobile App, so how to explain the consumers’ purchase intention?
- L318-L331 lists some literature, does this have anything to do with the performance of paid mobile Apps? What is the purpose of this part of the literature?
- L339 mentioned that “App usefulness refers to an individual’s belief”. The usefulness refers to a belief, such statement seems not accurate.
- Line 426, “The evaluation process indicated that the level of agreement among the members of the 426 panel was 87.5%.”, how to get 87.5%?
Reviewer 2 Report
This is a meaningful article. By using a mixed-methods approach, this article identifies the key factors that consumers consider in determining whether or not to purchase mobile Apps for their smartphones. An exploratory study involving a qualitative methods approach is done. The results show that seven decision factors have a direct impact on mobile App purchase intention. This work shows the results of the study.
The authors need to add more literature, such as "Determinants of the Intention to Adopt Mobile Augmented Reality Apps in Shopping Malls among University Students", "Measuring Ease of Use of Mobile Applications in E-commerce Retailing from the Perspective of Consumer Online Shopping Behaviour Patterns" and "Understanding Factors influencing Consumers Online Purchase intention Via Mobile App: Perceived Ease of use, Perceived Usefulness, System Quality, information Quality, and Service Quality".
Reviewer 3 Report
Title: Title contains important keywords, I suggest the author (s) make the title interesting, and attention-grabbing.
Comments on abstract and introduction:
The paper aims to study the key factors that consumers consider in determining whether or not to purchase mobile Apps for their smartphones.
It is noticed that the author (s) develops the research framework based on acceptable theories. In the abstract section, results are clearly addressed. The abstract is well organized.
Research gaps are explained, plenty of previous studies' findings are presented. However, I suggest the author (s) bold the originality and novelty of the paper. Importantly the significance of the study or the theory used that can influence existing issues should be discussed in terms of practice more clearly. Please notice that a component of the significance of the study is to elucidate exactly how your study will extend the current literature.
At the end of the introduction section, part authors explain that research contributes to mobile Apps research by advancing the understanding of decision factors with regard to mobile Apps purchases, and identifies the interrelationships among these factors by employing a mixed-methods approach. The results of this study can subsequently guide the efforts of developers and publishers as to how to generate revenue by stimulating Apps’ sales. I suggest the authors elaborate more on this part.
Comments on hypothesis development:
The theoretical background is well structured. Authors could logically and clearly present the ideas and motivations behind the proposed hypothesis. Key concepts are there. The authors have evaluated existing related studies. Assumptions and expectations are described. Authors still can elaborate more on relevant theories and models. The relationship between research constructs is mentioned.
Comments on research methodology:
The discussion on why the current method is suitable for the purpose of the current study is a clear way. Discussion on why sample size is pertinent is not evident.
Qualitative stages involved in setting questions and data collection are well explained.
PLS-SEM is also used for the purpose of the study.
Overall, the methodology part is well organized and clear.
Comments on results:
The author has responded to the proposed hypotheses. The result section is clear. Numbers and descriptive statistics are well presented. Please see Table 5 and explain why there is no VIF value for App purchase intention (INT)?
Comments on discussion and conclusions:
In the discussion, the part author has summarized the significance and purpose of the study. It is mentioned that seven decision factors have a direct impact on mobile App purchase intention. These factors include the price value of the App, App performance, App enjoyment, App trialability, electronic Word-of-Mouth (eWOM) about the App, App technical reliability, and App usefulness. The theoretical implications of the study are listed nicely, I expect brighter knowledge of practical and managerial contributions.
Round 2
Reviewer 1 Report
The author has made revisions in accordance with the reviewer’s comments. We are satisfied with the author’s revision and agree to accept